# OpenReview forum: "BigCodeBench: Benchmarking Code Generation with Diverse Function Calls and Complex Instructions"
_ICLR.cc/2025/Conference — ICLR 2025 Oral_

### Official Review · Reviewer_TG6s · 2024-11-01

**Soundness:** 4
**Presentation:** 4
**Contribution:** 4
**Rating:** 10
**Confidence:** 4

**Summary:**

This benchmark paper proposes improved execution-based evaluation for solving complex real-time application like code generation problems. It focuses on both the effective instruction query and the detailed test case setup for unit test sets, curated using human-LLM collaboration.

**Strengths:**

Quite a thorough analysis and structured approach of generation of benchmark is taken into account. With detailed literature review of past works in the field, the paper also demonstrates its effectiveness as an improved code generation benchmark and evaluation.

**Weaknesses:**

Apart from tiny bits of improvements needed, the paper is quite detailed in its analysis. However, it lacks the essence of what are the areas where these LLM are failing on the new benchmark, and a compared analysis of how an expert human would perform in the same. So as to note the scope of baseline improvement.

**Questions:**

- With respect to line 237, what were the reasons or examples of model failure in the dataset that was improved and captured by the annotators? Subsequently, what steps were taken to overcome it?
- In case of BigCodeBench-Instruct benchmark, each user query is structured well in terms of sequence and format of instruction for code generation. This might not be the case when it comes to user-like query. Analysis on part of robustness to ambiguity in NL, variation in structured format needs to be further analyzed and presented within the paper.
- Line 300, 'PL: Programming Language', is not used within context of the table and is unnecessary.
- Table 1: Tool Statistics, # Call column, format the column to have all '/' inline, like the adjacent columns.
- Perhaps, Figure 6 can be more visually summarized using scatter plot with grid colored based on size of parameter of LLM. Can be updated, if it helps provide better display of story.
- A definition of what calibrated scores should be added in description.
- Table 3 comparison can also be done against other datasets like MBPP Plus dataset, NaturalCodeBench and EvalPlus.
- Add on analysis of how new benchmark overcomes the issue of data contamination due to the benchmarks being used as training data in several LLM. What rephrasing or variation in query structuring is ensured to tackle this?
- Add in example of failure cases and categorize to why LLM still fails on new benchmark.
- How well does an expert perform comparatively?

---

> ### Author Response · Authors · 2024-11-16
> **Authors' Response (1/2)**
>
> Thank you for the constructive feedback! We appreciate your insights and would like to provide you with detailed explanations along with new experimental results to address your comments.
>
> > With respect to line 237, what were the reasons or examples of model failure in the dataset that was improved and captured by the annotators? Subsequently, what steps were taken to overcome it?
>
>
> One significant issue we observed was a misalignment between docstrings and test cases, leading to ambiguity in the generation programs. For example, a task might require models to return "Invalid url input" when an exception is raised while fetching data from an input URL. However, the corresponding test cases might expect "Invalid url input!" instead of "Invalid url input" for invalid URLs. This inconsistency causes models to fail such cases even when their outputs align with the original docstring. Since LLMs are not yet reliable enough to detect such misalignments automatically, annotators must manually review failures and update the docstrings to ensure consistency. To verify that models can correctly interpret the revised docstrings, we plan to validate the new outputs generated by GPT-3.5 against the updated test cases. This process aims to improve alignment and reduce ambiguity in future iterations.
>
>
> > In case of BigCodeBench-Instruct benchmark, each user query is structured well in terms of sequence and format of instruction for code generation. This might not be the case when it comes to user-like query. Analysis on part of robustness to ambiguity in NL, variation in structured format needs to be further analyzed and presented within the paper.
>
> Thank you for the great suggestions! We have added the following experiments to Appendix O.
>
> We would like to note that addressing ambiguous natural language (NL) instructions is not the primary focus of this work. However, investigating the robustness of LLMs in handling such ambiguities, as explored in [1], is certainly an interesting direction. Similar to our analysis of prompting techniques documented in Appendix K.3, we conducted preliminary experiments on several representative models using a paraphrased version of BigCodeBench-Hard. For rephrasing, we prompt Llama-3.1-8B-Instruct with ”Please rephrase the following programming description in clear and concise plain text. Keep the core meaning intact but use different wording. Write in a casual style from a user's perspective, without including any task completion markers.\n\n# Programming Description\n”. We set the temperature to 0.2 and top_k as 0.95.
>
> | Original Model                 | Original | Rephrased | Delta |
> |--------------------------------|----------|-----------|-------|
> | Qwen2.5-Coder-32B-Instruct     | 27.7     | 8.8       | -18.9 |
> | Llama-3.1-70B-Instruct         | 23.6     | 11.5      | -12.1 |
> | GPT-4o-mini-2024-07-18         | 23.6     | 10.1      | -13.5 |
> | GPT-4o-2024-05-13              | 25.0     | 12.8      | -12.2 |
>
> From the table, it is evident that Qwen2.5-Coder-32B-Instruct exhibits a more significant performance drop compared to other models, despite its stronger performance on the original BigCodeBench-Hard. While it is true that rephrased instructions may introduce additional ambiguity, the results indicate that some models may still lack sufficient robustness to handle less structured and more casual inputs. This highlights a potential area for future improvement in LLM capabilities, particularly in managing variations in natural language.
>
> > Table 1: Tool Statistics, # Call column, format the column to have all '/' inline, like the adjacent columns.
>
> We respectfully argue that this change may not enhance the presentation. The current table layout is intentionally designed so that the top table aligns with the bottom table, maintaining the same number of columns for consistency. The additional space between # and Cov. is a result of the context provided by the # Lib. and # Call. columns. This alignment ensures that the structure is uniform and easier to follow across both tables.
>
> > A definition of what calibrated scores should be added in description.
>
> We would like to note that the calibration setup is described in the first finding of Section 4.1. Specifically, on Lines 423–424, we mentioned that missing setup elements (e.g., import statements and global constants) were added back to the model outputs. We have highlighted the definition on Line 370 to ensure readers clearly understand the calibrated Pass@1.
>
> [1] Wang, S., Li, Z., Qian, H., Yang, C., Wang, Z., Shang, M., ... & Xiang, B. (2023, July). ReCode: Robustness Evaluation of Code Generation Models. In The 61st Annual Meeting Of The Association For Computational Linguistics.

---

> ### Author Response · Authors · 2024-11-16
> **Authors' Response (2/2)**
>
> > Table 3 comparison can also be done against other datasets like MBPP Plus dataset, NaturalCodeBench and EvalPlus.
>
> Thank you for the great suggestions! We have added the following experiments to Appendix N.
>
> | Dataset               | BigCodeBench-Complete |            | BigCodeBench-Instruct |            |
> |-----------------------|------------------------|------------|------------------------|------------|
> |                       | $r$          | $p$ | $r$          | $p$ |
> | MBPP+                 | 0.963                 | 0.937      | 0.926                 | 0.881      |
> | NaturalCodeBench      | 0.757                 | 0.803      | 0.913                 | 0.857      |
>
>
> As we can see from the table, MBPP+ is strongly correlated with BigCodeBench. The correlation between NaturalCodeBench (Python-English) and BigCodeBench-Complete is slightly lower, which is expected as NaturalCodeBench prompts are more similar to BigCodeBench-Instruct.
>
>
> > Add on analysis of how new benchmark overcomes the issue of data contamination due to the benchmarks being used as training data in several LLM. What rephrasing or variation in query structuring is ensured to tackle this?
>
> Thank you for raising this important question regarding data contamination and the steps taken to address it. We have carefully considered the potential risks of future data contamination, particularly the possibility of benchmarks being used as training data for LLMs. As discussed in our earlier response, we have taken initial steps to mitigate contamination by releasing BigCodeBench on Hugging Face instead of GitHub, as Hugging Face lacks the automated scraping mechanisms commonly associated with contamination from GitHub source code. However, we recognize that these measures alone cannot fully eliminate contamination risks, especially when high-quality benchmark data is integrated into closed-source models.
> To address this, we are developing a more generalized variant of BigCodeBench that replaces library API invocations with synthetically generated helper functions. This eliminates dependencies on specific libraries and allows for dynamic task generation within the same scope as BigCodeBench. By creating diverse tasks in this way, we aim to minimize contamination risks while maintaining the benchmark’s relevance for evaluating LLMs.
>
> > Add in example of failure cases and categorize to why LLM still fails on new benchmark.
>
> Please refer to the 10 examples and the explanations documented in Appendix L. Specifically, we find that LLMs like GPT-4o mainly have shallow understandings of specific APIs and tend to miss some logic in the generated programs. Meanwhile, we encourage the future studies on how LLMs misuse/mishandle APIs in the wild.
>
> > How well does an expert perform comparatively?
>
> As we noted in the last paragraph of Section 2.3 (Lines 249 - 252), the expert performance on the sampled tasks is 97%, far beyond the level of current LLMs. To improve the transparency of this work, we will consider adding the expert performance to the full/hard dataset.

---

> > ### Author Response · Authors · 2024-11-22
> > **Follow-up**
> >
> > Dear Reviewer,
> >
> > Please let us know if our response has addressed your concerns.
> >
> > Cheers

---

> > > ### Comment · Reviewer_TG6s · 2024-11-22
> > > **Thank You**
> > >
> > > Thanks for the timely and detailed responses. I'm really happy to see the additional efforts and findings you gathered along the way and that you will incorporate the additional items to highlight in the paper as well. It's really useful for the coding community. Also bumped the scores. Looking forward to the paper being at the conference!

---

### Official Review · Reviewer_pm7H · 2024-11-03

**Soundness:** 4
**Presentation:** 4
**Contribution:** 4
**Rating:** 10
**Confidence:** 3

**Summary:**

The authors present BigCodeBench, a massive high-quality benchmark to test Large Language Models (LLMs) ability to perform diverse function calls and solve complex code instructions. BigCodeBench comprises 1,140 fine-grained Python programming tasks after proper cleaning and filtering and covers functionality from 139 widely adopted programming libraries. Specifically, the benchmark evaluates two common programming scenarios (1) Code completion based on structured docstrings, and (2) the completion of programming tasks based on natural language instructions. The authors used powerful LLMs to source programming tasks, refactor code, and generate test cases, under constant human supervision. This construction process is a remarkable example of a perfect balance between LLM automation and human curation to ensure data quality and reliability. The authors conduct an extensive evaluation of state-of-the-art LLMs on BigCodeBench. Among others, the results show that LLMs are not yet capable of following complex instructions to use function calls precisely.

**Strengths:**

### Originality
The benchmark proves to be original for the following reasons:
- It contains diverse code, sourced from widely adopted Python libraries from seven domains.
- It contains more complex programming tasks than other benchmarks. Specifically, the authors report on task cyclomatic complexity compared to other function-level Python programming benchmarks, and a larger coverage of function calls compared to state-of-the-art function calling benchmarks.

### Quality

#### Benchmark Construction
The authors followed best practices to select, clean, and filter the seed data. They also integrate the use of LLMs into the process by keeping humans in the loop when needed. Moreover, code execution in a controlled environment is performed to ensure code quality.

#### Validation
- The authors extensively evaluate 60 state-of-the-art LLMs on BigCodeBench-Complete (code-completion based on docstrings) and 35 instruction-tuned LLMs on BigCodeBench-Instruct (programming tasks based on natural language instructions).

### Clarity
- The benchmark construction and evaluation processes are remarkably explained.
- Limitations of the dataset as well as future work are addressed in detail as part of the appendix.

### Significance
As BigCodeBench will be open-sourced it represents a significant contribution to the research community. Moreover, the study reveals interesting observations about the ability of LLMs in complex code-completion tasks, which sets a roadmap for the development of new training datasets to improve these capabilities.

**Weaknesses:**

Please fix the typo in line 182 (refractor -> refactor) to make even better this amazing work.

**Questions:**

- How do you distill observations such as "Interestingly, we observe that instruction-tuned LLMs can omit the essential import statements of the given prompts", do you rely on static analysis of automatic error logs? are humans also involved in the process?

---

> ### Author Response · Authors · 2024-11-16
> **Authors' Response**
>
> Thank you for recognizing our work!
>
> > How do you distill observations such as "Interestingly, we observe that instruction-tuned LLMs can omit the essential import statements of the given prompts", do you rely on static analysis of automatic error logs? are humans also involved in the process?
>
> We first noticed this issue in GPT-3.5-Turbo when validating the execution results. Then, we (as humans) observed more similar behaviors of other LLMs when we conducted more evaluations. Hope this makes sense :)

---

### Official Review · Reviewer_yub5 · 2024-11-04

**Soundness:** 3
**Presentation:** 4
**Contribution:** 4
**Rating:** 8
**Confidence:** 4

**Summary:**

This paper introduces a novel benchmark to evaluate the ability of cutting-edge LLMs in function calling and complex instruction following for code generation.

**Strengths:**

- The proposed benchmark introduces a challenging task that involves diverse function calls as tools and complex instruction-following for code generation. Current state-of-the-art LLMs achieve up to 60% Pass@1, while human performance reaches 97%, providing a clear benchmark for future research to improve on.
- By evaluating a wide range of both open-source and closed-source LLMs on this benchmark, the study establishes a strong foundation and adds credibility to the proposed framework.

**Weaknesses:**

- There is a risk of future data contamination [1, 2]. To mitigate this, it may be beneficial to release only the validation set while keeping the test set hidden, similar to practices used in NL2SQL benchmarks [3,4].

- Some domains, such as visualization tasks returning graph outputs, would benefit from examples illustrating how the test cases are constructed for evaluation.

[1] Yihong Dong, Xue Jiang, Huanyu Liu, Zhi Jin, Bin Gu, Mengfei Yang, and Ge Li. 2024. Generalization or Memorization: Data Contamination and Trustworthy Evaluation for Large Language Models. In Findings of the Association for Computational Linguistics: ACL 2024, pages 12039–12050, Bangkok, Thailand. Association for Computational Linguistics.

[2] Jain, Naman, et al. "Livecodebench: Holistic and contamination free evaluation of large language models for code." arXiv preprint arXiv:2403.07974 (2024).

[3] Tao Yu, Rui Zhang, Kai Yang, Michihiro Yasunaga, Dongxu Wang, Zifan Li, James Ma, Irene Li, Qingning Yao, Shanelle Roman, Zilin Zhang, and Dragomir Radev. 2018. Spider: A Large-Scale Human-Labeled Dataset for Complex and Cross-Domain Semantic Parsing and Text-to-SQL Task. In Proceedings of the 2018 Conference on Empirical Methods in Natural Language Processing, pages 3911–3921, Brussels, Belgium. Association for Computational Linguistics.

[4] Jinyang Li, Binyuan Hui, GE QU, Jiaxi Yang, Binhua Li, Bowen Li, Bailin Wang, Bowen Qin, Ruiying Geng, Nan Huo, Xuanhe Zhou, Chenhao Ma, Guoliang Li, Kevin Chang, Fei Huang, Reynold Cheng, & Yongbin Li (2023). Can LLM Already Serve as A Database Interface? A BIg Bench for Large-Scale Database Grounded Text-to-SQLs. In Thirty-seventh Conference on Neural Information Processing Systems Datasets and Benchmarks Track.

**Questions:**

- How to prevent future data contamination?
- How are test cases constructed to evaluate visualization tasks requiring graph outputs?

---

> ### Author Response · Authors · 2024-11-16
> **Authors' Response (1/2)**
>
> Thank you very much for your thorough and thoughtful review. We greatly appreciate the time and effort you have dedicated to providing detailed feedback. Below, we address your comments and concerns point by point.
>
> > How to prevent future data contamination?
>
> We completely agree with the suggestion to provide a hidden/private test set. However, we plan to make this part of the efforts in the next version of BigCodeBench. For the initial release, we believe that open-sourcing the BigCodeBench data offers significant benefits, as our primary motivation is to emphasize the importance of advancing the code generation capabilities of future LLMs. Our goal is not to create an incremental code generation benchmark with a similar scope to HumanEval or MBPP, but rather to push the boundaries in this domain.
>
> We have carefully considered potential future data contamination issues before releasing BigCodeBench. To mitigate this, we have decided to release the data on Hugging Face rather than directly on GitHub. Based on past experiences, most contamination stems from the unintentional inclusion of GitHub-sourced code, as seen with datasets like HumanEval and MBPP. Unlike GitHub, Hugging Face does not support the kind of automated scraping that typically leads to contamination, making BigCodeBench relatively safer from this issue.
>
> That said, we acknowledge that it is impossible to ensure complete privacy of benchmark data. For instance, when closed-source model APIs are used for inference, companies may collect and use the data for training if it is deemed high-quality. Preventing this would require access to their model weights and the ability to run them locally, which is not feasible in most cases.
> In addition, we are developing a more generalized variant of BigCodeBench that replaces library API invocations with synthetically generated helper functions. This eliminates dependencies on specific libraries and allows for dynamic task generation within the same scope as BigCodeBench. By creating diverse tasks in this way, we aim to minimize contamination risks while maintaining the benchmark’s relevance for evaluating LLMs.

---

> > ### Comment · Reviewer_yub5 · 2024-11-22
> > **Reviewer yub5's Response**
> >
> > Thank you for your response.
> >
> > > We have carefully considered potential future data contamination issues before releasing BigCodeBench. To mitigate this, we have decided to release the data on Hugging Face rather than directly on GitHub.
> >
> > Although I think Hugging Face is also very accessible (in fact, it might be more accessible than building a crawler), I still understand the authors' point.
> >
> > > For instance, when closed-source model APIs are used for inference, companies may collect and use the data for training if it is deemed high-quality. Preventing this would require access to their model weights and the ability to run them locally, which is not feasible in most cases.
> >
> > I agree with this, but my point was about only providing test problems without releasing ground truth code snippets and full test cases, as NL2SQL benchmarks like [BIRD](https://bird-bench.github.io/) have done. The test evaluation is done only through submitting generated code to the official organization.

---

> > > ### Author Response · Authors · 2024-11-23
> > >
> > > Thank you for getting back to us!
> > >
> > > > my point was about only providing test problems without releasing ground truth code snippets and full test cases, as NL2SQL benchmarks like BIRD have done. The test evaluation is done only through submitting generated code to the official organization.
> > >
> > > At this stage, we believe maintaining transparency is essential to gather valuable community feedback for further refinement. We will begin constructing a synthetic private dataset once any model achieves an average score exceeding 50% on BigCodeBench-Hard or 60% on BigCodeBench, whichever occurs first.

---

> > > > ### Author Response · Authors · 2024-11-25
> > > >
> > > > Dear Reviewer,
> > > >
> > > > As the deadline for the discussion period approaches, we kindly request your feedback on the recent response. Your insights would be greatly appreciated.
> > > >
> > > > Thank you for your time and consideration!

---

> ### Author Response · Authors · 2024-11-16
> **Authors' Response (2/2)**
>
> > How are test cases constructed to evaluate visualization tasks requiring graph outputs?
>
> We attach the test cases for a visualization programming task, where the function `task_func` returns both a DataFrame and a bar chart Axes object for validation. The DataFrame's `Category` and `Value` columns are tested to ensure they correctly represent the input data. The Axes object is validated for its title (`"Category vs Value"`) and bar properties, including heights and x-axis labels, using a helper method, `is_bar`. The test cases cover diverse inputs with varying data sizes and values, ensuring the function reliably handles both data transformation and visualization requirements. We note that the test case design for the visualization tasks is similar to DS-1000 [1] but with more detailed validation.
>
> We have updated Appendix M to document this example.
> ```
> import pandas as pd
> import matplotlib.pyplot as plt
> import seaborn as sns
>
>
> def task_func(list_of_pairs):
>     """
>     Create a Pandas DataFrame from a list of pairs and visualize the data using a bar chart.
>     - The title of the barplot should be set to 'Category vs Value'`.
>
>     Parameters:
>     list_of_pairs (list of tuple): Each tuple contains:
>         - str: Category name.
>         - int: Associated value.
>
>     Returns:
>     tuple:
>         - DataFrame: A pandas DataFrame with columns 'Category' and 'Value'.
>         - Axes: A matplotlib Axes displaying a bar chart of categories vs. values.
>
>     Requirements:
>     - pandas
>     - matplotlib.pyplot
>     - seaborn
>
>     Example:
>     >>> list_of_pairs = [('Fruits', 5), ('Vegetables', 9)]
>     >>> df, ax = task_func(list_of_pairs)
>     >>> print(df)
>          Category  Value
>     0      Fruits      5
>     1  Vegetables      9
>     """
>     pass
>
>
> import unittest
> class TestCases(unittest.TestCase):
>     """Test cases for the task_func function."""
>     @staticmethod
>     def is_bar(ax, expected_values, expected_categories):
>         extracted_values = [
>             bar.get_height() for bar in ax.patches
>         ]  # extract bar height
>         extracted_categories = [
>             tick.get_text() for tick in ax.get_xticklabels()
>         ]  # extract category label
>         for actual_value, expected_value in zip(extracted_values, expected_values):
>             assert (
>                 actual_value == expected_value
>             ), f"Expected value '{expected_value}', but got '{actual_value}'"
>         for actual_category, expected_category in zip(
>             extracted_categories, expected_categories
>         ):
>             assert (
>                 actual_category == expected_category
>             ), f"Expected category '{expected_category}', but got '{actual_category}'"
>     def test_case_1(self):
>         df, ax = task_func(
>             [
>                 ("Allison", 49),
>                 ("Cassidy", 72),
>                 ("Jamie", -74),
>                 ("Randy", -25),
>                 ("Joshua", -85),
>             ]
>         )
>         # Testing the DataFrame
>         self.assertEqual(
>             df["Category"].tolist(), ["Allison", "Cassidy", "Jamie", "Randy", "Joshua"]
>         )
>         self.assertEqual(df["Value"].tolist(), [49, 72, -74, -25, -85])
>         # Testing the plot title
>         self.assertEqual(ax.get_title(), "Category vs Value")
>         self.is_bar(
>             ax=ax,
>             expected_categories=["Allison", "Cassidy", "Jamie", "Randy", "Joshua"],
>             expected_values=[49, 72, -74, -25, -85],
>         )
>
>     # More test cases...
> ```
>
> [1] Lai, Y., Li, C., Wang, Y., Zhang, T., Zhong, R., Zettlemoyer, L., ... & Yu, T. (2023, July). DS-1000: A natural and reliable benchmark for data science code generation. In International Conference on Machine Learning (pp. 18319-18345). PMLR.

---

> > ### Author Response · Authors · 2024-11-22
> > **Follow-up**
> >
> > Dear Reviewer,
> >
> > Please let us know if our response has addressed your concerns.
> >
> > Cheers

---

### Official Review · Reviewer_9WoB · 2024-11-06

**Soundness:** 3
**Presentation:** 3
**Contribution:** 3
**Rating:** 8
**Confidence:** 4

**Summary:**

This paper introduces BigCodeBench, a novel code generation benchmark that improves existing ones like HumanEval by incorporating more diverse function calls and complex instructions. BigCodeBench comprises 1,140 Python code generation tasks, necessitating the use of 139 libraries across 7 domains. The dataset was constructed through a semi-automated framework leveraging collaboration between LLMs and human experts. In this process, LLMs are prompted to generate coding tasks based on sampled code snippets, performing both initial generation and refinement of programs and test cases, while human annotators continuously provide feedback to enhance quality. Extensive experimental evaluations reveal that current LLMs struggle with accurately executing complex instructions and function calls, achieving scores of up to 60%, significantly below the human performance of 97%. These results highlight the ongoing challenges and the need for further advancements in the field of code generation.

**Strengths:**

- The paper makes a valuable contribution to code generation benchmarks by incorporating diverse function calls and complex instructions, offering a novel perspective on evaluating LLM capabilities in library and tool use. Results indicate that even top-performing LLMs face challenges with these tasks, highlighting opportunities for further advancements.
- The paper is well-structured and easy to follow. The benchmark curation framework is technically sound, back with a systematic design and thorough quality checks. The extensive evaluation provides some insights into how current LLMs handle library usage and follow complex instructions.

**Weaknesses:**

- The technical contribution of the benchmark curation framework appears incremental, as the collaborative approach between LLMs and human is already widely adopted in LLM research.
- While Python libraries are essential for task solutions in this benchmark, the rapid evolution and versioning of these libraries can introduce compatibility issues. The authors should consider methods to mitigate the effects of version incompatibility in the evaluation process.

**Questions:**

- The Python libraries used in this study are mostly popular ones, likely present in the pretraining data of most LLMs. Future work could explore how LLMs adapt to newer, domain-specific libraries that may not have been included in their training data.
- Based on the benchmark results, what potential approaches could improve the performance of existing LLMs in handling complex instructions and function calls? Insights on targeted training techniques or model adjustments would be valuable.

---

> ### Author Response · Authors · 2024-11-16
> **Authors' Response (1/2)**
>
> Thanks a lot for your valuable review. We would like to address your concerns as follows:
>
> > The technical contribution of the benchmark curation framework appears incremental, as the collaborative approach between LLMs and human is already widely adopted in LLM research.
>
> We respectfully argue that the proposed construction pipeline has its own merit, given that this is specifically designed for code generation. In addition, our project has already commenced over a year ago, specifically in May of last year, as documented in Appendix O. At that time, there was limited work focusing on LLM-based data synthesis for code (Step 1) and agentic workflows for code refinement and test case generation (Step 2). While we recognize that these workflows could independently stand as contributions prior to the development of the BigCodeBench dataset, we believe they are more appropriately presented as integral components of our comprehensive data construction pipeline. Publishing them together in this manner offers a more coherent narrative and emphasizes the synergy between these steps. As there is **almost no detailed documentation on collaborative code data construction** in the prior work, we strongly believe that transparently documenting this extensive data pipeline within a single paper will provide significant value to the research community. By doing so, we aim to offer a holistic view of the framework’s design and its practical utility. We sincerely hope you will reconsider the merits of this integrated approach.
>
> > While Python libraries are essential for task solutions in this benchmark, the rapid evolution and versioning of these libraries can introduce compatibility issues. The authors should consider methods to mitigate the effects of version incompatibility in the evaluation process.
>
> Thank you for your thoughtful suggestion! We share similar concerns and have discussed the limitations in Appendix F.1.
> To make the future evaluation more reliable, we can consider adding the library metadata (e.g., versions) when prompting the models. We suggest that the truly capable LLM should be able to determine the specific APIs available in the given versions. To aid such evaluations, we have documented the metadata in Appendix H.3.
> Additionally, we are happy to offer our thoughts on mitigating and improving the benchmark data concerning the issue of API evolution, as outlined below:
> We believe that leveraging LLM-based API updates represents a promising direction for exploration. However, this approach entails several challenges from a software engineering perspective. **First**, Python APIs can evolve in numerous ways. A study of 288 releases across six popular Python libraries [1] identifies 14 distinct types of API evolution (e.g., class removal, method removal, parameter reordering, and field addition). Notably, 11 of these types are considered breaking changes, directly impacting program behaviour. Addressing API removal is particularly complex, as evolved APIs may lack direct replacements in newer library versions. **Second**, as highlighted by [2], a significant number of deprecated APIs are inadequately documented, posing further challenges for developers seeking to address deprecated usage.
> To address these issues, we propose a hybrid approach that combines static program analysis with multi-turn LLM interaction. Specifically, we suggest utilizing static program analysis to compare APIs across different library versions and employing LLMs to identify potential changes, including those not explicitly documented. Rule-based methods can then analyze whether programming tasks in BigCodeBench involve deprecated APIs. For tasks requiring updates, LLMs can leverage the new library documentation and identified API changes to propose updated ground-truth solutions and revise deprecated APIs in test cases.
> For cases involving API removal, where replacements are unavailable, LLMs may require additional interaction rounds to generate valid implementations. Automatic validation of these updated solutions and test cases can be achieved by grounding LLMs within a code sandbox for multi-turn execution.
> We believe this approach provides a practical pathway to mitigate the challenges associated with API evolution, and we look forward to further exploring and refining these ideas.
>
> [1] Zhang, Z., Zhu, H., Wen, M., Tao, Y., Liu, Y., & Xiong, Y. (2020, February). How do python framework apis evolve? an exploratory study. In 2020 ieee 27th international conference on software analysis, evolution and reengineering (saner) (pp. 81-92). IEEE.
>
> [2] Wang, J., Li, L., Liu, K., & Cai, H. (2020, November). Exploring how deprecated python library apis are (not) handled. In Proceedings of the 28th acm joint meeting on european software engineering conference and symposium on the foundations of software engineering (pp. 233-244).

---

> ### Author Response · Authors · 2024-11-16
> **Authors' Response (2/2)**
>
> > The Python libraries used in this study are mostly popular ones, likely present in the pretraining data of most LLMs. Future work could explore how LLMs adapt to newer, domain-specific libraries that may not have been included in their training data.
>
> We fully agree with this suggestion, which is also partially addressed in Appendix F.1. The primary reason for selecting popular libraries is that they align more closely with real-world software development practices and offer maximum benefit to the community. We note that some concurrent works like SciCode [3] help address the concern of lacking domain-specific libraries. Additionally, given the limited number of domain experts involved in data annotation, there is a risk that they may lack sufficient familiarity with the latest or domain-specific libraries. This could lead to errors or ambiguities in annotating programming tasks that involve such libraries. Moreover, emerging libraries often lack maturity and tend to introduce breaking changes, making it challenging to construct programming tasks with stable APIs. By focusing on widely used and well-established libraries, we aim to ensure a more reliable and impactful dataset for the community.
>
>
> > Based on the benchmark results, what potential approaches could improve the performance of existing LLMs in handling complex instructions and function calls? Insights on targeted training techniques or model adjustments would be valuable.
>
> Thank you for your interest in improving the performance of current LLMs. We are actively working on developing a more generalized benchmark, building upon BigCodeBench-Hard, where all APIs and partial logic are replaced with hand-written functions. Based on our observations from comparing this new benchmark with the original BigCodeBench, we have identified two major limitations in current LLMs: (1) Inaccurate API Understanding: LLMs often fail to precisely interpret APIs, including their input and output types, leading to incorrect API invocations, and (2) Insufficient Fine-Grained Instruction Following: LLMs struggle with adhering to detailed, domain-specific instructions (e.g., in code), frequently overlooking critical aspects such as case-by-case exception handling.
> We believe the first issue can be partially addressed through multi-turn environment interaction, which has the potential to significantly enhance LLMs' comprehension of APIs. Regarding the second issue, current datasets for code instruction tuning, such as Magicoder-OSS-Instruct [4], tend to be relatively high-level, which limits their effectiveness in enabling LLMs to follow detailed instructions. To address this, we suggest training on synthetic fine-grained instructions, similar in granularity to BigCodeBench, as a promising approach to enhance the ability to handle complex, precise instructions.
>
> While these strategies may help mitigate surface-level issues, we also believe that the Process-Supervised Reward Model (PRM) [5] presents a more promising long-term solution for improving code generation quality. However, PRM comes with its own challenge: it requires a substantial amount of high-quality, step-level labelled data, which is non-trivial to obtain.
>
> We are excited to explore these directions further and welcome additional insights or collaborations to address these challenges effectively.
>
> [3] Tian, M., Gao, L., Zhang, S. D., Chen, X., Fan, C., Guo, X., ... & Peng, H. (2024). Scicode: A research coding benchmark curated by scientists. arXiv preprint arXiv:2407.13168.
>
> [4] Wei, Y., Wang, Z., Liu, J., Ding, Y., & Zhang, L. (2024). Magicoder: Empowering code generation with oss-instruct. In Forty-first International Conference on Machine Learning.
>
> [5] Lightman, H., Kosaraju, V., Burda, Y., Edwards, H., Baker, B., Lee, T., ... & Cobbe, K. (2023). Let's verify step by step. arXiv preprint arXiv:2305.20050.

---

> ### Author Response · Authors · 2024-11-22
> **Follow-up**
>
> Dear Reviewer,
>
> Please let us know if our response has addressed your concerns.
>
> Cheers

---

> > ### Author Response · Authors · 2024-11-24
> >
> > Dear Reviewer,
> >
> > As the deadline for the discussion period approaches, we kindly request your feedback on our response. Your insights would be greatly appreciated.
> >
> > Thank you for your time and consideration!

---

> > > ### Comment · Reviewer_9WoB · 2024-11-27
> > >
> > > Thanks for the comprehensive response that mostly resolves my concern. I have increased my rating accordingly.

---

### Author Response · Authors · 2024-11-16

We thank all reviewers for their detailed reviews. We have made the following updates to the paper with the additional blue text.

- **Replacement**: Correction to “refractor” on Line 182, as pointed out by Reviewer pm7H.
- **Removal**: PL description on Line 300, as pointed out by Reviewer TG6s.
- **Addition**: Definition of calibrated Pass@1 on Line 370, as asked by Reviewer TG6s.
- **Addition**: Thoughts on future data contamination in Appendix C, as asked by Reviewers yub5 and TG6s.
- **Addition**: Discussions on library diversity of BigCodeBench in Appendix F.1, as asked by Reviewer 9WoB.
- **Addition**: Design of BigCodeBench-Evolved to mitigate evolved APIs in Appendix F.4, as asked by Reviewer 9WoB.
- **Addition**: Design of test cases in BigCodeBench for visualization tasks in Appendix M, as asked by Reviewer yub5.
- **Addition**: Correlation comparisons to more programming benchmarks in Appendix N, as asked by Reviewer TG6s.
- **Addition**: Evaluation on less-structured instruction prompts in Appendix O, as asked by Reviewer TG6s.

Furthermore, we sincerely ask the reviewers to check out the Appendix for more details, though it might be a bit lengthy. It may address some of your concerns already.

We appreciate the positive sentiment expressed about BigCodeBench, and are very excited about future work utilizing and building on BigCodeBench. We will continue to maintain the benchmark and the leaderboard.

---

### Meta-Review · Area_Chair_WEDe · 2024-12-19

**Metareview:**

The paper presents a new code generation benchmark for evaluating large language models. Overall, the benchmark is comprehensive and more challenging with diverse function calls and complex instructions as opposed to previous generation of benchmarks like HumanEval. I believe this will be a useful contribution to the community. I will request the authors to consider making it as easy possible though. A point about "compatibility issues with future python versions" was mentioned in the review which I feel is critical for widespread use of the benchmark.

**Additional Comments On Reviewer Discussion:**

Most reviewers liked the paper and had concerns about benchmark usage and data contamination which were addressed in the rebuttal.

---

### Decision · Program_Chairs · 2025-01-22

Accept (Oral)